# Facile Synthesis of Ni_x_Co_3−x_S_4_ Microspheres for High-Performance Supercapacitors and Alkaline Aqueous Rechargeable NiCo-Zn Batteries

**DOI:** 10.3390/nano12172994

**Published:** 2022-08-30

**Authors:** Daojun Zhang, Bei Jiang, Chengxiang Li, Hao Bian, Yang Liu, Yingping Bu, Renchun Zhang, Jingchao Zhang

**Affiliations:** 1Henan Key Laboratory of New Optoelectronic Functional Materials, College of Chemistry and Chemical Engineering, Anyang Normal University, Anyang 455000, China; 2College of Chemistry, Zhengzhou University, Zhengzhou 450001, China

**Keywords:** Ni_x_Co_3−x_S_4_ structures, micro/nanostructures, supercapacitors, Ni-Zn batteries

## Abstract

Electrochemical energy storage devices (EESDs) have caused widespread concern, ascribed to the increasing depletion of traditional fossil energy and environmental pollution. In recent years, nickel cobalt bimetallic sulfides have been regarded as the most attractive electrode materials for super-performance EESDs due to their relatively low cost and multiple electrochemical reaction sites. In this work, NiCo-bimetallic sulfide Ni_x_Co_3−x_S_4_ particles were synthesized in a mixed solvent system with different proportion of Ni and Co salts added. In order to improve the electrochemical performance of optimized Ni_2.5_Co_0.5_S_4_ electrode, the Ni_2.5_Co_0.5_S_4_ particles were annealed at 350 °C for 60 min (denoted as Ni_2.5_Co_0.5_S_4_-350), and the capacity and rate performance of Ni_2.5_Co_0.5_S_4_-350 was greatly improved. An aqueous NiCo-Zn battery was assembled by utilizing Ni_2.5_Co_0.5_S_4_-350 pressed onto Ni form as cathode and commercial Zn sheet as anode. The NiCo-Zn battery based on Ni_2.5_Co_0.5_S_4_-350 cathode electrode delivers a high specific capacity of 232 mAh g^−1^ at 1 A g^−1^ and satisfactory cycling performance (65% capacity retention after 1000 repeated cycles at 8 A g^−1^). The as-assembled NiCo-Zn battery deliver a high specific energy of 394.6 Wh kg^−1^ and long-term cycling ability. The results suggest that Ni_2.5_Co_0.5_S_4_-350 electrode has possible applications in the field of alkaline aqueous rechargeable electrochemical energy storage devices for supercapacitor and NiCo-Zn battery.

## 1. Introduction

In the past ten years, rechargeable electrochemical energy storage devices (EESDs) have attracted extensive attention all over the world due to the increasing energy crisis and environmental pollution [1,2,3,4,5]. At present, lithium-ion batteries dominate the world markets in energy storage field due to their long lifetime. However, the theoretical capacity of lithium-ion batteries is low, and they still have safety issues, which cannot meet the growing demand of high-performance energy storage [6,7]. Recently, transition-metal sulfides (TMSs) with different compositions, morphologies, and structures have been developed and demonstrate excellent performances in various applications [8,9,10,11,12,13,14,15,16]. Supercapacitors and Ni-Zn batteries, as typical alkaline aqueous EESDs, have some advantages: low cost, safety, and quickly charge-discharge speed. Thus, supercapacitors and Ni-Zn batteries demonstrate great potential applications in the rechargeable clean energy storage field [17,18].

So far, Co and Ni-based metal sulfides have been synthesized and used as electrode materials for alkaline aqueous energy storage devices such as supercapacitors and aqueous Ni-Zn batteries [19,20,21,22,23,24,25]. Nickel cobalt bimetallic sulfides such as NiCo_2_S_4_, [26,27,28] CoNi_2_S_4_, [29,30] and Ni_x_Co_3−x_S_4_, [31] are regarded to be the most attractive electrode materials for EESDs due to their low cost and synergistic electrochemical reaction sites. For instance, Wang and his collaborators demonstrated a MOF-derived NiCo_2_S_4_ and carbon hybrid hollow spheres compactly concatenated by electrospun carbon nanofibers as a binder-free cathode electrode for NiCo_2_S_4_/HCS@CFs//Zn battery, and the constructed battery exhibited good performance with high capacity (343.1 mA h g^−1^ at 3.8 A g^−1^) and superior rate performance [32]. Wang et al. used a facile two-step solution-based method to synthesize 3D interconnected NiCo_2_S_4_ nanosheets integrated into nickel foam, and the constructed NiCo_2_S_4_/nickel foam electrode for supercapacitors delivers a high areal specific capacitance of 10.82 F cm^−2^ at 10 mA cm^−2^ [33]. Hierarchically hollow C/NiCo_2_S_4_ nanosphere composites were developed by Xing using SiO_2_ nanosphere and as the hard template, and the as-obtained hollow C/NiCo_2_S_4_ nanosphere as supercapacitor electrode exhibits an extraordinary specific capacitance of 1545 F g^−1^ at 2 A g^−1^ and enhanced cycling stability [34].

Lou group developed a facile self-templating conversion way to synthesize Ni_x_Co_3−x_S_4_ hollow prisms, and the hollow structure used as supercapacitors electrode revealed excellent pseudo-capacitive performance with high specific capacitance (895.2 F g^−1^ at 1 A g^−1^) and cycling stability [35]. However, single component Ni_x_Co_3−x_S_4_ electrodes face great challenge in achieving high energy density, high power density, and superior rate performance. In order to overcome this obstacle, constructing a composite sulfide structure is an efficient method. 

Inspired by the above reported research of nickel cobalt bimetallic sulfide electrode materials, in this work, a facile solvothermal method with water-ethanol-oleamine system [36] was utilized to fabricate porous structured Ni_x_Co_3−x_S_4_ (x = 0.9, 1.8, 2.5) with different nickel cobalt ratios, denoted as Ni_0.9_Co_2.1_S_4_, Ni_1.8_Co_1.2_S_4_, and Ni_2.5_Co_0.5_S_4_, respectively. The as-prepared Ni_2.5_Co_0.5_S_4_ electrode exhibits the highest specific capacitance among the three samples. In order to obtain an optimal sample with high specific capacitance, we used it as a precursor to construct composite sulfide structure. Furthermore, the capacitance and rate performance of Ni_2.5_Co_0.5_S_4_ electrode was greatly enhanced after annealing at 350 °C for 60 min (denoted as Ni_2.5_Co_0.5_S_4_-350). The fantastic synergistic effect between NiS and CoNi_2_S_4_ in Ni_2.5_Co_0.5_S_4_-350 led to enhanced electrochemical performance, and the Ni_2.5_Co_0.5_S_4_-350 with composite sulfide structure displayed a high specific capacitance of 2001 F g^−1^ at a current density of 1 A g^−1^. In addition, as a cathode for an aqueous NiCo-Zn battery, the Ni_2.5_Co_0.5_S_4_-350 electrode demonstrates a high specific capacity of 232 mAh g^−^^1^ at 1 A g^−1^ and excellent long-time cycling performance. Moreover, the as-assembled NiCo-Zn battery delivers a high specific energy of 394.6 Whkg^−^^1^ and long-term cycling ability.

## 2. Experimental Section

### 2.1. Chemicals

CoCl_2_·6H_2_O, NiCl_2_·6H_2_O, Zn(Ac)_2_·2H_2_O, L-cysteine, N-methyl-2-pyrrolidone (NMP), Ni foam, zinc foil, ethanol, and potassium hydroxide (KOH) were purchased from Sinopharm Chemical Reagent Co., Ltd. (Shanghai, China). Oleamine and acetylene black were purchased from Aladdin Industrial Corporation (Shanghai, China). All chemicals were used without further purification in this work.

### 2.2. Preparation of Ni_x_Co_3−x_S_4_ Microspheres

At first, 0.1M CoCl_2_·6H_2_O and NiCl_2_·6H_2_O solution were prepared and stored for later use. To prepare Ni_2.5_Co_0.5_S_4_ particles, as a typical preparation process, 0.81g oleamine and 4 mL ethanol were added to a 20 mL autoclave under strong stirring, then the as-prepared CoCl_2_·6H_2_O (0.1 mL) and NiCl_2_·6H_2_O (0.4 mL) solution, 0.5 mL H_2_O and 0.0485g L-cysteine, were added and further stirred for 30 min to form a uniform solution. Next, the autoclave was sealed and heated at 180 °C for 12 h. The black precipitate was obtained and washed with ethanol three times. For the synthesis of Ni_1.8_Co_1.2_S_4_ and Ni_0.9_Co_2.1_S_4_ samples, the synthesis process is the same as that of Ni_2.5_Co_0.5_S_4_, except for the difference of added metal salts solution (0.165 mL CoCl_2_·6H_2_O and 0.335 mL NiCl_2_·6H_2_O for Ni_1.8_Co_1.2_S_4_, 0.335 mL CoCl_2_·6H_2_O, and 0.165 mL NiCl_2_·6H_2_O for Ni_0.9_Co_2.1_S_4_). In order to improve the electrochemical performance of Ni_2.5_Co_0.5_S_4_ electrode, the Ni_2.5_Co_0.5_S_4_ particles were annealed at 350 °C for 60 min and denoted as Ni_2.5_Co_0.5_S_4_-350.

### 2.3. Material Characterization

The phase of the as-synthesized Ni_x_Co_3−x_S_4_ samples was checked via X-ray diffraction (XRD) on a PANalytical X’ Pert. The surface morphology and microstructure were studied on SEM (Hitachi SU8010, Japan) and TEM (FEI Tecnai G2, USA) equipment, respectively. The energy-dispersive X-ray spectroscopy (EDS) and X-ray photoelectron spectroscopy (ESCALab 250XI, USA) were used to analyze composition and chemical states of the as-synthesized samples.

### 2.4. Electrode Preparation and Electrochemical Measurements

The supercapacitor positive electrode was prepared by manual coating method, and the electrode slurry was firstly prepared via mixing Ni_x_Co_3−x_S_4_, acetylene black, and polyvinylidene-fluoride (the mass ratio is 8:1:1) with 110 µL of N-methylpyrrolidone. Then, the slurry was cautiously coated onto a piece of cleaned Ni foam (1cm × 1cm) and dried at 80 °C in a vacuum for one night. The investigation of Ni-Zn battery in a two-electrode cell was carried out on an electrochemical workstation (CHI 660E) using the Ni_2.5_Co_0.5_S_4_-350 coated Ni form as positive electrode and a commercial Zn plate as negative electrode, respectively, and the mixed solution of 2 M KOH and 0.2 M Zn(Ac)_2_·2H_2_O was used as electrolytes. The cycle voltammetry (CV) and galvanostatic charge/discharge (GCD) for supercapacitor and Ni-Zn battery were also performed on CHI 660E. The Nyquist plots were obtained at open circuit potential with the frequency range of 0.01Hz-100KHz. The calculation formula [18,21,37] of specific capacitance (*C*_s_, F g^−1^), specific capacity (*C*_m_, mAh g^−1^), energy density (*E*, Wh kg^−1^), and power density (*P*, kW kg^−1^) are provided in the supporting information.

## 3. Results and Discussion

Figure 1 illustrates the preparation of porous Ni_x_Co_3−x_S_4_ microspheres via a mixed solvothermal method in a water–ethanol–oleamine solvent system. First, the morphology and microstructure of the series Ni_x_Co_3−x_S_4_ (x = 0.9, 1.8, 2.5) particles were detected by scanning electron microscopy (SEM) and transmission electron microscopy (TEM). Figure 1a–c illustrates SEM images of the as-prepared Ni_0.9_Co_2.1_S_4_, Ni_1.8_Co_1.2_S_4_, and Ni_2.5_Co_0.5_S_4_ microsphere samples, respectively. SEM images show the uniform microsphere morphology with smooth surface, the average diameters for as-prepared Ni_0.9_Co_2.1_S_4_ microspheres is ~1.93 µm, after doping high contents of Ni ions, the corresponding particles sizes increase from 2.26 to 2.46 µm (Appendix A). The energy-dispersive X-ray spectrum (EDS) results demonstrated the Co and Ni elements successfully included in the series sulfides samples. EDS element mapping clearly confirmed Co, Ni and S elements are uniform distribution in the Ni_x_Co_3−x_S_4_ samples (Figure 1d–f). The atomic percentage (at.%) of Co/Ni/S in Ni_2.5_Co_0.5_S_4_ is 39.58: 7.01: 53.41, and the ratio is fitted to the stoichiometry of Ni_2.5_Co_0.5_S_4_. The Ni/Co/S at.% of Ni_0.9_Co_2.1_S_4_ and Ni_1.8_Co_1.2_S_4_ samples is 15.04: 35.80: 49.16 and 31.46: 19.15: 49.39, respectively (Figure 1h–i, Appendix A), which was consistent with the feed molar ratios of Co and Ni elements.

In addition, Ni_2.5_Co_0.5_S_4_ sample was annealed at 350 °C in the muffle furnace for 60 min and denoted as Ni_2.5_Co_0.5_S_4_-350. After the annealing treatment, the appearance of the Ni_2.5_Co_0.5_S_4_ was almost not changed, although their size decreased and the surface became somewhat rough. The typical Ni_2.5_Co_0.5_S_4_-350 microsphere structure was further investigated by SEM and TEM, and the corresponding SEM and TEM images are shown in Figure 2a–c. The HAADF image and EDS elemental mapping images shown in Figure 2e indicated that the Co, Ni, and S elements were uniformly distributed in Ni_2.5_Co_0.5_S_4_-350 microsphere. The observed lattice interplanar fringe in the core of Ni_2.5_Co_0.5_S_4_-350 microsphere is ~0.33 nm (marked in grey box), which corresponded well to the (022) plane of cubic CoNi_2_S_4_ (Figure 2f). Furthermore, the interplanar spacing of the outer layer of Ni_2.5_Co_0.5_S_4_-350 microsphere is 0.29 nm, which indicated the existence of NiS after annealing process. 

The phase composition and crystallinity of as-synthesized microparticles were further checked by X-ray power diffraction (XRD). Figure 3a exhibited the typical XRD patterns for the three microspheres samples obtained by one step solvothermal process, and all the diffraction peaks appearing in the patterns match well with the siegenite CoNi_2_S_4_ (JCPDS No.: 96-900-9853). The diffraction peaks of Ni_2.5_Co_0.5_S_4_ at 16.12, 26.57, 31.30, 37.91, 50.04, and 54.83° can be respectively assigned to (111), (022), (113), (004), (115), and (044) planes of cubic CoNi_2_S_4_ phase. The diffraction peaks of more Co ions content in the microspheres shift to higher angle of 2 theta in the patterns. Figure 3b visibly exhibited the XRD pattern of thermal treatment Ni_2.5_Co_0.5_S_4_-350 microsphere. Except for the prominent peaks of CoNi_2_S_4_, the diffraction peaks at 30.0, 34.5, 45.7, and 53.3° can be ascribed to hexagonal NiS phase, indicating that the composite sulfide structure was successfully synthesized via the calcination process at 350 ℃. The above result is consisted with the HRTEM characterization shown in Figure 2f.

In addition, compared with Ni_2.5_Co_0.5_S_4_, the annealed microsphere is conducive to supply larger specific surface area, and the corresponding isothermal curves are shown in Figure 3c,d. The four samples all showed typical IV type isotherms (Figure 3c and Appendix A). In Figure 3c, the hysteresis loop at the relative pressure ranged from 0.5 to 0.9, which indicated the presence of mesoporous structure. In Figure 3d, the trend of steep increase in the range of 0.6~1.0 indicated the abundant meso-macroporous structure of the annealed sample. The calculated BET surface area of Ni_2.5_Co_0.5_S_4_ and Ni_2.5_Co_0.5_S_4_-350 samples is 3.98 and 9.11 m^2^ g^−1^, respectively, which is smaller than that of the Ni_0.9_Co_2.1_S_4_ and Ni_1.8_Co_1.2_S_4_ samples (Appendix A). The main pore size of the Ni_2.5_Co_0.5_S_4_ is between 3-7 nm and Ni_2.5_Co_0.5_S_4_-350 samples showed a wide distribution, which further confirmed that the meso-macropores formed after the calcined process (Insert in Figure 3c,d). The increase of specific surface area of Ni_2.5_Co_0.5_S_4_-350 not only provides abundant active sites for Faradic reaction but also enhances the mass transportation between the electrolyte and electrode. 

The XPS survey of Co_0.5_Ni_2.5_S_4_ microspheres and its sintered derivative Ni_2.5_Co_0.5_S_4_-350 is shown in Figure 4a, which indicates the existence of Ni, Co, S, O, C, and traces of N elements. The presence of N 1s may result from the surface absorption of oleamine during the synthesis process. In Figure 4b, the Co 2p high resolution spectrum of Co_0.5_Ni_2.5_S_4_ microspheres can be divided to two double peaks centered at 782.0/798.6 eV and 778.8/793.8 eV, attributed to Co^2+^ and Co^3+^, respectively. The fitted Co^2+^ peaks of calcined microspheres (Co_0.5_Ni_2.5_S_4_-350) display a dramatic increase, while the integral area of commensurable Co^3+^ peaks reduce compared with that for Co_0.5_Ni_2.5_S_4_. The high-resolution spectrum of Ni 2p demonstrate that the dominant peaks at 856.9 and 875.3 eV significantly boost after the annealing process, characteristic of higher content of Ni^3+^ in annealed microspheres than that of Co_0.5_Ni_2.5_S_4_ (Figure 4c). The S 2p spectrum was deconvoluted to two groups of peaks located at 161.3/162.5 eV and 161.6/163.9 eV, which is assigned to Co-S and Ni-S bonds, respectively, indicating that Co and Ni ions coexist in the homologous sulfide, whereas the satellite peak at 169.4 eV might be ascribed to SO_4_^2−^, due to possible partial surface oxidation of the sulfide exposed in air (Figure 4d).

The Ni 2p_3/2_ peaks of Co_0.5_Ni_2.5_S_4_-350 display a positive shift of ~0.4 eV compared with Co_0.5_Ni_2.5_S_4_ sample, indicating the strong interaction between NiS and CoNi_2_S_4_ obtained by the annealing process. Simultaneously, the Co 2p peaks also present a negative shift after the calcination process, which may be attributed to the transfer of electron from NiS to CoNi_2_S_4_ at the interface of Co_0.5_Ni_2.5_S_4_-350 microspheres. The Ni-S and Co-S also exhibit some shifts, which further confirmed the strong electron interaction at the newly formed composite interface of Co_0.5_Ni_2.5_S_4_-350 sample. Thus, this phenomenon will help to enhance the electrochemical performance of Co_0.5_Ni_2.5_S_4_ -350 sample [34,38].

To evaluate the as-synthesized Ni_x_Co_3−x_S_4_ (x = 0.9, 1.8, 2.5) samples as electrode materials in rechargeable EESDs, the electrochemical supercapacitor tests were conducted by a standard three-electrode system in aqueous solution (2.0 M KOH). The cyclic voltammetry (CV) curves of the Ni_0.9_Co_2.1_S_4_, Ni_1.8_Co_1.2_S_4_, and Ni_2.5_Co_0.5_S_4_ electrodes were performed within a potential window of 0 to 0.5 V and exhibited in Figure 5a and Appendix A. Two obvious redox peaks located at 0.34/0.15 V were observed in the CV curves for all the Ni_x_Co_3−x_S_4_ samples. The maximum nickel doped sample of Ni_2.5_Co_0.5_S_4_ possessed the largest integrated area and peak intensity. In general, as the electrode capacitance is proportional to integrated areas of CV curves, the Ni_2.5_Co_0.5_S_4_ sample can deliver the largest pseudocapacitive capacity. The GCD curves of the three Ni_x_Co_3−x_S_4_ electrodes at a current density of 2 A g^−1^ within a potential window of 0–0.42 V were studied and compared in Figure 5b. Figure 5c depicted the specific capacitances of the as-prepared Ni_x_Co_3−x_S_4_ particles constructed electrodes calculated from GCD curves (Appendix A) at various current densities (2-10 A g^−1^). Specific capacitances of the Ni_2.5_Co_0.5_S_4_ electrode were calculated to be 1335, 1161, 1042, 925, and 846 F g^−1^ at current densities of 2, 4, 6, 8, 10 A g^−1^, respectively. The Ni_2.5_Co_0.5_S_4_ electrode still maintains 63.3% of initial capacitance at 10 A g^−1^. In comparison, Ni_0.9_Co_2.1_S_4_ and Ni_1.8_Co_1.2_S_4_ electrodes also display relatively high specific capacitances of 959 and 1259 F g^−1^ at the current density of 2 A g^−1^, and at the high current density of 10 A g^−1^, the capacitance of the two electrodes decreased to 698 and 555 F g^−1^ with the corresponding capacitance retentions of 72.8% and 44.1%, respectively. The long-term cycling stability test of the Ni_x_Co_3−x_S_4_ electrodes at 4 A g^−1^ is exhibited in Figure 5d. The capacitance of Ni_2.5_Co_0.5_S_4_ electrode keeps 50.0% of the initial value after 3000 repeated cycles. It is worth mentioning that Ni_0.9_Co_2.1_S_4_ exhibited significantly high cycle stability of 101% initial capacity after long-term cycling.

Oleamine can exist on surface of the particles via coordination bond action and may block some electrochemical sites to reduce the energy storge performance. Thus, the Ni_2.5_Co_0.5_S_4_ electrode material was located through high temperature treatment at 350 °C and denoted as Ni_2.5_Co_0.5_S_4_-350. Furthermore, the capacitance and rate performance of Ni_2.5_Co_0.5_S_4_-350 electrode was greatly enhanced compared with Ni_2.5_Co_0.5_S_4_. The CV and GCD curves shown in Figure 6a,b, Ni_2.5_Co_0.5_S_4_-350 electrode displayed a larger enclosed area of CV curve and longer discharge time of GCD curve compared to the other three Ni_x_Co_3−x_S_4_ electrodes, which indicated the superior charge storage ability and capacitive performance of the Ni_2.5_Co_0.5_S_4_-350 electrode. The Ni_2.5_Co_0.5_S_4_-350 electrode displays a high specific capacitance of 2001 F g^−1^ at a current density of 1 A g^−1^, good rate capability (1795@10Ag^−1^, 89.7% capacitance retention, Figure 6c), and satisfactory cycling performance (69% of the initial value after 1500 repeated cycles, Figure 6d). The morphology of the Ni_2.5_Co_0.5_S_4_-350 electrode after the cycling test has little change compared with that before cycling (Appendix A). Appendix A exhibited the Nyquist plots of the as-prepared series electrodes. The inherent resistance (*R*_s_) can be detected from the intercept at real axis, the *R*_s_ of Ni_0.9_Co_2.1_S_4_, Ni_1.8_Co_1.2_S_4_, and Ni_2.5_Co_0.5_S_4_-350 are similar (~1.0 Ω), which are all lower than that of Ni_2.5_Co_0.5_S_4_ electrode (1.52 Ω). The Ni_2.5_Co_0.5_S_4_-350 electrode owns the lowest charge transfer resistance (*R*_ct_) among the four electrodes. In the low frequency region, the Ni_2.5_Co_0.5_S_4_-350 electrode also shows the steepest slope with the smallest Warburg impedance (*Z*_w_) of 2.03 Ω (Appendix A), indicating the lowest diffusion resistance of Ni_2.5_Co_0.5_S_4_-350 electrode obtained via annealing process. The lowest electro-transfer and fast ion diffusion resistance of Ni_2.5_Co_0.5_S_4_-350 leads to its improved electrochemical performance [39,40]. The significantly enhanced performance can be comparable to the many reported Co-Ni bimetal sulfides electrodes such as hollow C/NiCo_2_S_4_ nanosphere (1545 F g^−1^ at 2 A g^−1^) [34], carbon-containing NiCo_2_S_4_ hollow-nanoflake electrode (1722 F g^−1^ at 1 A g^−1^) [41], onion-like NiCo_2_S_4_ electrode (1016 F g^−1^ at 2 A g^−1^) [42], NiCo_2_S_4_ hollow spheres (756 F g^−1^ at 1 A g^−1^) [43], nitrogen-doped carbon nanofibers@ NiCo_2_S_4_ composite (1078 F g^−1^ at 1 A g^−1^) [44], eggplant-derived carbon@NiCo_2_S_4_ (1394.5 F g^−1^ at 1 A g^−1^) [45], amorphous CoNi_2_S_4_ nanocages (1890 F g^−1^ at 4 A g^−1^) [46], and double-shelled zinc-cobalt sulfide (Zn-Co-S) rhombic dodecahedral cages (1266 F g^−1^ at 1 A g^−1^) [47]. The detailed supercapacitor performances of Ni_2.5_Co_0.5_S_4_-350 compared with the related sulfides electrodes are listed in Appendix A.

Thus, Ni_2.5_Co_0.5_S_4_-350 electrode was further explored in in-depth studies in aqueous alkaline Zn ion batteries, a Ni_2.5_Co_0.5_S_4_-350//Zn battery was fabricated by using the as-fabricated Ni_2.5_Co_0.5_S_4_-350 as cathode and commercial Zn plate as anode, respectively, and the mixed solution of 2 M KOH and 0.2 M Zn(Ac)_2_ was used as an electrolyte. The cathode and anode reaction in an alkaline solution is expressed as follows [22,23]: Ni-Co-S +2OH^−^ → NiSOH + CoSOH + 2e^−^
(1)
CoSOH +OH^−^ → CoS(OH)_2_ + e^−^
(2)
[Zn(OH)_4_]^2−^ + 2e^−^ → Zn + 4 OH^−^
(3)

Overall reaction equation
[Zn(OH)_4_]^2−^+ 2e^−^ + Ni-Co-S→ NiSOH + CoS(OH)_2_ + Zn (4)

Figure 7a depicts the CV curves of Ni_2.5_Co_0.5_S_4_-350//Zn battery at different scan rates (1-10 mV s^−1^) with a wide voltage range of 1.2-1.95 V vs. Zn. The Ni_2.5_Co_0.5_S_4_-350//Zn exhibits a pair of obvious redox peaks at 1.85 and 1.65 V at 1 mV s^−1^, and the symmetric redox peaks can still be preserved at 10 mV s^−1^. The GCD of Ni_2.5_Co_0.5_S_4_-350//Zn at various current densities of 1-10 A g^−1^ were shown in Figure 7b, as clearly observed from the GCD curves, and the charge and discharge plateau are approximately 1.79 and 1.70 V (1 A g^−1^), respectively. At a high current density of 10 A g^−1^, the charging/discharging plateau are still stable, and the discharging voltage plateau decreases slowly with increasing current densities from 1 to 10 A g^−1^. At the current densities of 1, 2, 4, 6, 8, and 10 A g^−1^, the Ni_2.5_Co_0.5_S_4_-350//Zn battery yields a high specific capacity of 232, 217, 202, 191, 181, and 172 mAh g^−1^, respectively. With the 10-times increasement of current density, the calculated capacity retention of 74.1% indicates the good rate performance of Ni_2.5_Co_0.5_S_4_-350//Zn battery in alkaline solution. In addition, when the discharge current density switches to 1 A g^−1^ after 60 cycles, the battery exhibits specific capacity nearly the same as the initial value, illustrating the high rate capability and stable reversibility of the battery. 

The specific capacity of constructed Ni_2.5_Co_0.5_S_4_-350//Zn battery can beat several recently reported aqueous Ni-Zn batteries, such as NiCo_2_O_4_//Zn (222.7 mAh g^−1^) [48], NiCo_2_O_4_@ CC//Zn (183.1 mAh g^−1^ at 1.6 A g^−1^) [49], Ni_2_P/C //Zn (176 mAh g^−1^ at 1 A g^−1^) [50], Co_3_O_4_@NF//Zn (162 mAh g^−1^ at 1 A g^−1^) [51], Zn//NiO–CNTs battery (155 mAh g^−1^ at 1 A g^−1^) [52], and Ni_3_S_2_//Zn (148 mAh g^−1^) [53]. Furthermore, long-term cycling stability was also evaluated. After 1000 repeated cycles at a high current of 8 A g^−1^, the Ni_2.5_Co_0.5_S_4_-350//Zn battery still retains 54% capacity retention and has satisfactory cycling stability. 

The capacitive and diffusion contribution percentage of Ni_2.5_Co_0.5_S_4_-350 electrode at varied scan rates is shown in Figure 8a, and it is clearly seen that the capacitive contribution increases with the increase of scan rate. Diffusion effect is negatively correlated to scan rate. Power and energy density values of Ni_2.5_Co_0.5_S_4_-350//Zn battery were calculated from capacity values obtained at various current densities and the resulted Ragone plot is shown in Figure 8b. The Ni_2.5_Co_0.5_S_4_-350//Zn battery achieved the energy density of 394.6 W h kg^−1^ at 1.7 kW kg^−1^. The energy storage performance of Ni_2.5_Co_0.5_S_4_-350//Zn battery is comparable with the previously reported Ni-Zn batteries such as Ni_2_P/C//Zn (318 Wh kg^−1^ at 1.376 kW kg^−1^) [50], β-Ni(OH)_2_/CNFs// Zn (325 Wh kg^−1^ at 1.23 kW kg^−1^) [54], Ni-Co_9_S_8_-0.6//Zn (256.5 Wh kg^−1^ at 1.69 kW kg^−1^) [55], DBS-NiCo_2_O_4_//Zn (326.5 Wh kg^−1^ at 0.822 kW kg^−1^) [56], P-Co_3_O_4_//Zn battery (193.7 Wh kg^−1^ at 1.6 kW kg^−1^) [57]. NiCo_2_S_4_@NiMoO_4_/Ni_2_P//Zn (384 Wh kg^−1^ at 0.46 kW kg^−1^) [58], Ni/NiO-BCF //Zn (313.4 Wh kg^−1^ at 0. 66 kW kg^−1^) [59], NiCo-S-2/RGO//Zn (333.2 Wh kg^−1^ at 1.7 kW kg^−1^) [60] (Appendix A). The acquired outstanding specific capacity and high energy density, along with the facile preparation method and low-cost, endows the as-prepared Ni_2.5_Co_0.5_S_4_-350 electrode with great potential applications in aqueous electrochemical energy storage devices.

## 4. Conclusions

In summary, a series of bimetallic sulfide Ni_x_Co_3−x_S_4_ particles were synthesized via a simple solvothermal method, and the electrochemical performances as electrode materials in alkaline aqueous rechargeable EESDs, including supercapacitor and NiCo-Zn battery, were carefully studied. As a cathode for an aqueous NiCo-Zn battery, the Ni_2.5_Co_0.5_S_4_-350 electrode demonstrates a high specific capacity of 232 mAh g^−^^1^ at 1 A g^−1^ and excellent long-time cycling performance at a high current density of 8 A g^−1^. Moreover, NiCo-Zn battery also emerges with a high energy density of 394.6 W h kg^−1^ and a power density of 1.7 kW kg^−^^1^. The acquired good electrochemical energy storage properties might be ascribed to their multiple electrochemical reaction sites and high electrical conductivity. The achieved outstanding specific capacity and high energy density of Ni_2.5_Co_0.5_S_4_-350 electrode endows it with great potential applications in aqueous electrochemical energy storage devices.

## Data Availability

Not applicable.

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
