# Peer review of "Facile Synthesis of NixCo3−xS4 Microspheres for High-Performance Supercapacitors and Alkaline Aqueous Rechargeable NiCo-Zn Batteries"

_nanomaterials, 2022, doi:10.3390/nano12172994_

Round 1
Reviewer 1 Report
Facile synthesis of NixCo3-xS4 microspheres for high-performance supercapacitors and alkaline aqueous rechargeable NiCo-Zn batteries
The study reports on the synthesis of NixCo3-xS4 particles using mixed solvent systems and their application in supercaps and Ni-Zn batteries. In spite of the fact that Ni-Zn is an old battery, the results on particle synthesis are good.
I am listing here a few related papers (not my papers) that should be discuss in detail, for example:
1. Cheng, Ye, et al. "Facile synthesis of NixCo3-xS4 hollow nanoprism with broader electromagnetic absorption properties: Effect of Ni/Co atomic ratios." Journal of Alloys and Compounds 767 (2018): 323-329.
2. Jia, Zirui, et al. "Fabrication of NixCo3-xS4 hollow nanosphere as wideband electromagnetic absorber at thin matched thickness." Ceramics International 45.13 (2019): 15854-15859.
General comments:
1. Good manuscript especially on properties of NixCo3-xS4 particles.
Title:
1. Good
Abstract:
1. Abstract is too long. The rest is good
2. Remove connecting words such as herein, furthermore, respectively, and moreover. Concise and packed abstracts are best.
Introduction:
1. Lines 47-74 of paragraph 2 are too long, make two paragraphs. I am having a difficult time finding the problem statement, in view of the fact that NixCo3-xS4 has been synthesized for quite a while, what is your problem statement for this compound?
2. The article objective should be written at the end of the introduction, “In this works…”
3. Introduction should at least 3 components/paragraphs:
a. introduction to the field
b. problem statement
c. Objective and method
Experimental:
1. Brand name for CoCl2, NiCl2 etc
2. What is 3 electrode-system for CV? And how about the cell system for GCD?
3. SCE is not suitable for alkaline electrolyte.
Results and Discussion:
1. Figure 1: An explanation of how to calculate EDX results (based on atomic ratios), does it match with Ni0.9Co2.1S4, Ni1.8Co1.2S4, and 126 Ni2.5Co0.5S4
2. Figure 2, 3: TEM and XRD, explain about plane
3. It is better to write several paragraphs rather than one long paragraph.
Conclusion
1. Need to be answered your objectives and problem statements
2. Too long
Refs
1. References must be within five years (2017-2021)
Author Response
Comments and Suggestions for Authors
Facile synthesis of NixCo3-xS4 microspheres for high-performance supercapacitors and alkaline aqueous rechargeable NiCo-Zn batteries
The study reports on the synthesis of NixCo3-xS4 particles using mixed solvent systems and their application in supercaps and Ni-Zn batteries. In spite of the fact that Ni-Zn is an old battery, the results on particle synthesis are good.
I am listing here a few related papers (not my papers) that should be discuss in detail, for example:
- Cheng, Ye, et al. "Facile synthesis of NixCo3-xS4 hollow nanoprism with broader electromagnetic absorption properties: Effect of Ni/Co atomic ratios." Journal of Alloys and Compounds 767 (2018): 323-329.
- Jia, Zirui, et al. "Fabrication of NixCo3-xS4 hollow nanosphere as wideband electromagnetic absorber at thin matched thickness." Ceramics International 45.13 (2019): 15854-15859.
Response: The above articles have been cited in the revised version (Ref.15 and 16).
General comments:
- Good manuscript especially on properties of NixCo3-xS4 particles.
Title:
- Good
Abstract:
- Abstract is too long. The rest is good
Response: The Abstract was simplified in the revised manuscript.
- Remove connecting words such as herein, furthermore, respectively, and moreover. Concise and packed abstracts are best.
Response: The connecting words such as herein, furthermore, respectively, and moreover were removed in the revised manuscript.
Introduction:
- Lines 47-74 of paragraph 2 are too long, make two paragraphs. I am having a difficult time finding the problem statement, in view of the fact that NixCo3-xS4 has been synthesized for quite a while, what is your problem statement for this compound?
Response: Lines 47-74 of paragraph 2 of Introduction section was divided into two paragraphs in the revised version.
- The article objective should be written at the end of the introduction, “In this works…”
Response: The article objective has been added at the end of the introduction.
- Introduction should at least 3 components/paragraphs:
- introduction to the field
- problem statement
- Objective and method
Response: Introduction has been revised for 4 components/paragraphs in the revised version.
Experimental:
- Brand name for CoCl2, NiCl2 etc
Response: Brand name for all chemical reagents used in this work (CoCl2·6H2O, NiCl2·6H2O, etc) were added in the revised version (2.1 Chemicals)
- What is 3 electrode-system for CV? And how about the cell system for GCD?
Response: The supercapacitor performances including cyclic voltammetry (CV) and galvanostatic charge–discharge (GCD) were tested in a typical three electrode system, The NixCo3-xS4 coated Ni form was used as working electrode, Hg/Hg2Cl2 (SCE) electrode and Pt foil was used as the reference electrode and counter electrode, respectively.
The investigation of Ni-Zn battery in a two-electrode cell using the Ni2.5Co0.5S4-350 coated Ni form as positive electrode and a commercial Zn plate as negative electrode, respectively.
- SCE is not suitable for alkaline electrolyte.
Response: In the literature, SCE is also used under alkaline conditions. The concentration of KOH (2M) used in this work is not very high and the alkalinity is not very strong.
Results and Discussion:
- Figure 1: An explanation of how to calculate EDX results (based on atomic ratios), does it match with Ni9Co2.1S4, Ni1.8Co1.2S4, and 126 Ni2.5Co0.5S4
Response: We added the explanation of how to calculate EDX results and exhibited in Table S1 in the revised supporting information.
- Figure 2, 3: TEM and XRD, explain about plane
Response: The planes in TEM and XRD have been explained in the revised manuscript.
- It is better to write several paragraphs Results and Discussion.
Response: We write several paragraphs to replace one long paragraph for Results and Discussion.
Conclusion
- Need to be answered your objectives and problem statements
Response: Our objectives and problem statements have been added in the revised version.
- Too long
Response: In the revised manuscript, we have streamlined the Conclusion part.
Refs
- References must be within five years (2017-2021)
Response: References have been changed in the revised version.

Reviewer 2 Report
The authors prepared the NiCo-bimetallic sulfide NixCo3-xS4 particles in a mixed solvent system with different Ni and Co salts added. The as-prepared Ni2.5Co0.5S4 particles exhibit a specific capacity of 1335 F g−1 at a current density of 2 A g−1. Furthermore, an aqueous NiCo-Zn battery was assembled by utilizing Ni2.5Co0.5S4-350 pressed onto Ni form as cathode and commercial Zn sheet as an anode, respectively. The NiCo-Zn battery based on Ni2.5Co0.5S4-350 cathode electrode delivers a specific capacity of 232 mAh g−1 at 1 A g −1. This article can be re-consider after major revisions as below:
1. Line 17,
2. The work’s novelty should be elaborated on in the introduction section’s last paragraph.
3. Figure2 (e) related EDS, spectra, and the wt% and elemental % table need to be added.
4. The Ni2.5Co0.5S4 particles were annealed at 350 °C for 60 min, in the presence of which gas?
5. Fig. 3 (a) and (b) XRD, the plane of the respective peaks need to be labeled in the figure.
6. What type of JCPDS number is this: JCPDS 96-900-9853? In the figure also need to write JCPDS before the number. But this seems to be a cubic NiCo2S4 phase (COD: 96-900-9853 or JCPDS No. 43–1477) with space group Fd3m(227) and lattice parameter a= 9.4240 Å. Please be sure about that from a proper literature survey.
7. Figures 3 (c) and (d) need more explanation.
8. In figure 4 (a), the authors must write Co0.5Ni2.5S4-350 in red font.
9. The shift in the peak positions of Ni2p, Co2P, and S2P before and after annealing at 350 must be explained clearly.
10. The data presentation style explanation can be referenced from the following article with a proper citation; Journal of Electroanalytical Chemistry, 856, 2020, 113670 and ACS Appl. Energy Mater. 2021, 4, 1, 404–415.
11. The sample naming will be more scientific if the authors write the sample name as NixCo3-xS4, where x=0.9,1.8,2.5.
12. Many more research articles about NixCo3-xS4 material for energy applications are already reported in various journals; what makes this work different from others?
13. The authors need to compare the supercapacitor data and the Zn-Air batteries data of this material to the similar type of NixCo3-xS4 material by making a comparison table.
14. Specific capacity and specific capacitance are two different terms; here, the authors calculate the specific capacitance of the supercapacitor in this article. Hence, specific capacitance needs to replace the specific capacity of articles and figures. But for the Zn -batteries, it is a specific capacity.
15. Why the capacitance retention of the Ni0.9Co2.1S4 is better than others, whereas the CV and GCD of Ni2.5Co0.5S4 are better?
16. Why are the potential Window of the CV and GCDare not the same?
17. EIS of all electrode materials needs to be included.
18. The formulas used for calculation need to add in the main article; (can be referenced from the Carbon,179, July 2021, 89-99 and Chemical Engineering Journal,450, 5, 2022, 138363 with proper citation) and calculation methods related to the supercapacitor's specific capacitance and the Zn battery's specific capacity need clear explanations in the supporting information file. I doubt the mistakes in the calculations.
19. I wonder "The as-prepared Ni2.5Co0.5S4 particles exhibit a high specific capacitance of 1335 F g−1 at a current density of 2 A g−1, but the authors also show the Ni2.5Co0.5S4-350 electrode display a high specific capacitance of 2001 F g−1 at a current density of 1 A g−1,(1795@10A g−1)" why they claim the 1335 F g−1 at a current density of 2 A g−1 is best in the abstract and other????? It needs to be revised and the cv and the GCD of the Ni2.5Co0.5S4-350 need to compare in figure 5 (a) and (b).
20. First 3 samples GCD of 1 A g-1 was not taken why? It must be taken and compared with the Ni2.5Co0.5S4-350.
21. The Ni2.5Co0.5S4-350 electrode display high specific capacitance of 2001 F g−1 at a current density of 1 A g−1, the rate capability (1795@10A g−1, 80% capacitance retention, Fig.6c). But ((1795/2001 )*100%)=89.7% retention. Due to such type of blunders in the articles, we ask you to show the formulas and calculations in the ESI file. ( Follow comment number 18)
22. In Line 257 to 261, the equation font need to be revised, and the proper citation needs to be referenced.
23. The characterization of the sample after the stability test needs to be added. (Fe-SEM and EDX with color mapping).
Author Response
Comments and Suggestions for Authors
The authors prepared the NiCo-bimetallic sulfide NixCo3-xS4 particles in a mixed solvent system with different Ni and Co salts added. The as-prepared Ni2.5Co0.5S4 particles exhibit a specific capacity of 1335 F g−1 at a current density of 2 A g−1. Furthermore, an aqueous NiCo-Zn battery was assembled by utilizing Ni2.5Co0.5S4-350 pressed onto Ni form as cathode and commercial Zn sheet as an anode, respectively. The NiCo-Zn battery based on Ni2.5Co0.5S4-350 cathode electrode delivers a specific capacity of 232 mAh g−1 at 1 A g −1. This article can be re-consider after major revisions as below:
- Line 17,
Response: Revised.
- The work’s novelty should be elaborated on in the introduction section’s last paragraph.
Response: The work’s novelty was elaborated on in the introduction section’s last paragraph in the revised version.
- Figure2 (e) related EDS, spectra, and the wt% and elemental % table need to be added.
Response: The at.% and elemental % table has been added in the revised supporting information as Table S1.
- The Ni2.5Co0.5S4 particles were annealed at 350 °C for 60 min, in the presence of which gas?
Response: The Ni2.5Co0.5S4 particles were annealed at 350 °C for 60 min under air condition.
- Fig. 3 (a) and (b) XRD, the plane of the respective peaks need to be labeled in the figure.
Response: The plane of the respective peaks has been labeled in the Fig. 3 (a) and (b) in the revised version.
- What type of JCPDS number is this: JCPDS 96-900-9853? In the figure also need to write JCPDS before the number. But this seems to be a cubic NiCo2S4 phase (COD: 96-900-9853 or JCPDS No. 43–1477) with space group Fd3m (227) and lattice parameter a= 9.4240 Å. Please be sure about that from a proper literature survey.
Response: We added the JCPDS before the number in the Fig. 3 (a) and (b) in the revised version, and we checked the JCPDS No. 96-900-9853 represent CoNi2S4, which matched well with the NixCo3-xS4 series samples with high Ni content. While, the JCPDS No. 43–1477 is assigned to NiCo2S4 with high content of Co, which is not suitable for our synthesized samples.
- Figures 3 (c) and (d) need more explanation.
Response: Figures 3 (c) and (d) were fully explained in the revised manuscript.
- In figure 4 (a), the authors must write Co0.5Ni2.5S4-350 in red font.
Response: Co0.5Ni2.5S4-350 was changed in red font (Fig. 4, Fig.6, Fig.7 and Fig.8b) in the revised manuscript.
- The shift in the peak positions of Ni2p, Co2P, and S2P before and after annealing at 350 must be explained clearly.
Response: The shift in the peak positions of Ni 2p, Co 2p, and S 2p before and after annealing at 350 have been explained clearly in the revised manuscript. The Ni 2p3/2 peaks of Co0.5Ni2.5S4-350 display a positive shift of ~0.4 eV compared with Co0.5Ni2.5S4 sample, indicating the strong interaction between NiS and CoNi2S4 obtained by annealing process. Simultaneously, the Co 2p peaks also present a negative shift after calcination process, which may attribute to the transfer of electron from NiS to CoNi2S4 at the interface of Co0.5Ni2.5S4-350 microspheres. The Ni-S and Co-S also exhibit some shifts, which further confirmed the strong electron interaction at the newly formed composite interface of Co0.5Ni2.5S4-350 sample. Thus, this phenomenon will help to enhance the electrochemical performance of Co0.5Ni2.5S4 -350 sample.
- The data presentation style explanation can be referenced from the following article with a proper citation; Journal of Electroanalytical Chemistry, 856, 2020, 113670 and ACS Appl. Energy Mater. 2021, 4, 1, 404–415.
Response: The above articles have been cited in the revised version (Ref. 39 and 40).
- The sample naming will be more scientific if the authors write the sample name as NixCo3-xS4, where x=0.9,1.8,2.5.
Response: The samples were denoted as NixCo3-xS4 (x=0.9, 1.8, 2.5) in the revised version.
- Many more research articles about NixCo3-xS4 material for energy applications are already reported in various journals; what makes this work different from others?
Response: We have added relevant explanations in the revised version.
- The authors need to compare the supercapacitor data and the Zn-Air batteries data of this material to the similar type of NixCo3-xS4 material by making a comparison table.
Response: We compared the supercapacitor and the Zn-ions battery data of this material to the similar type of NixCo3-xS4 materials and listed in a Table S3 and Table S4 in the revised supporting information.
- Specific capacity and specific capacitance are two different terms; here, the authors calculate the specific capacitance of the supercapacitor in this article. Hence, specific capacitance needs to replace the specific capacity of articles and figures. But for the Zn-batteries, it is a specific capacity.
Response: The expression of specific capacitance of supercapacitor and specific capacity of Zn ion battery were revised in the articles and figures.
- Why the capacitance retention of the Ni0.9Co2.1S4 is better than others, whereas the CV and GCD of Ni2.5Co0.5S4 are better?
Response: The capacitance retention of Ni0.9Co2.1S4 is better than others may be due to its smallest charge transfer resistance (0.60 Ω) and largest specific surface area among the three NixCo3-xS4 (x=0.9, 1.8, 2.5) electrodes (Fig.S5, Table S2 and Fig.S2). The CV and GCD of Ni2.5Co0.5S4 are better than the other two electrodes, because the increase of Ni content are beneficial in improving the specific capacitance of Ni-Co sulfides ().
- Why are the potential Window of the CV and GCD are not the same?
Response: We selected potential window of the CV in a wide range, and potential window in the GCD curves cannot reach as CV curves.
- EIS of all electrode materials needs to be included.
Response: EIS of all electrode materials were added and analyzed in the revised version in Fig.S5. Fig.S5 exhibited the Nyquist plots of the as-prepared series electrodes. The inherent resistance (Rs) can be detected from the intercept at real axis, the Rs of Ni0.9Co2.1S4, Ni1.8Co1.2S4, and Ni2.5Co0.5S4-350 are similar (~1.0 Ω), which all lower than that of Ni2.5Co0.5S4 electrode (1.52 Ω). The Ni2.5Co0.5S4-350 electrode owns the lowest charge transfer resistance (Rct) among the four electrodes. In low frequency region, the Ni2.5Co0.5S4-350 electrode also shows the steepest slope with the smallest Warburg impedance (Zw) of 2.03 Ω, indicating the lowest diffusion resistance of Ni2.5Co0.5S4-350 electrode obtained via annealing process. The lowest electro-transfer and fast ion diffusion resistance of Ni2.5Co0.5S4-350 leads to its improved electrochemical performance. The corresponding Rs, Rct, and Zw value were listed in Table S2.
- The formulas used for calculation need to add in the main article; (can be referenced from the Carbon,179, July 2021, 89-99 and Chemical Engineering Journal,450, 5, 2022, 138363 with proper citation) and calculation methods related to the supercapacitor's specific capacitance and the Zn battery's specific capacity need clear explanations in the supporting information file. I doubt the mistakes in the calculations.
Response: The above articles have been cited in the revised version (Ref.18 and 37). The formulas used for calculation was add in the revised supporting information as calculation section.
The specific capacitance of NixCo3-xS4 (x=0.9, 1.8, 2.5) electrode using as supercapacitor can be calculated by
Where, m represents the mass of active material (g), I,Δt,ΔV is current (A), discharge time (s), and discharge potential window, respectively.
The specific capacity (Cm), energy density (E) and power density (P) of Ni2.5Co0.5S4-350//Zn battery were calculated as follows:
Where m, I, andΔt is the mass of active material in cathode (g), applied discharge current (A), and discharge time (h), respectively.
Where Cm (mAh g-1), V (V) represent the specific capacity of Ni2.5Co0.5S4-350//Zn battery and the corresponding discharge plateau.
- I wonder "The as-prepared Ni2.5Co0.5S4 particles exhibit a high specific capacitance of 1335 F g−1 at a current density of 2 A g−1, but the authors also show the Ni2.5Co0.5S4-350 electrode display a high specific capacitanceof 2001 F g−1 at a current density of 1 A g−1, (1795@10A g−1)" why they claim the 1335 F g−1 at a current density of 2 A g−1 is best in the abstract and other? It needs to be revised and the cv and the GCD of the Ni2.5Co0.5S4-350 need to compare in figure 5 (a) and (b).
Response: The claim of "the as-prepared Ni2.5Co0.5S4 particles exhibit a high specific capacitance of 1335 F g−1 at a current density of 2 A g−1 is best" in the abstract was deleted in the revised version. Among the Ni0.9Co2.1S4, Ni1.8Co1.2S4, and Ni2.5Co0.5S4 electrode, only Ni0.9Co2.1S4 of the three samples can be easily charged to 0.42 V at 1 A g−1. Therefore, we provided the comparison of GCD curves for all NixCo3-xS4 (x=0.9, 1.8, 2.5) samples at 2 A g−1 (Fig.5b). This phenomenon has also been reported in similar literature.
- First 3 samples GCD of 1 A g−1was not taken why? It must be taken and compared with the Ni2.5Co0.5S4-350.
Response: Among the NixCo3-xS4 series electrodes, only Ni0.9Co2.1S4 electrode can be easily charged to 0.42 V at 1 A g−1. All the as-prepared electrodes can be easily charged to 0.42 V at 2 A g−1, and we have compared the specific capacitance at 2 A g−1 for NixCo3-xS4 series electrodes and Ni2.5Co0.5S4-350 electrode in Table S3.
- The Ni2.5Co0.5S4-350 electrode display high specific capacitance of 2001 F g−1 at a current density of 1 A g−1, the rate capability (1795@10A g−1, 80% capacitance retention, Fig.6c). But ((1795/2001)*100%)=89.7% retention.Due to such type of blunders in the articles, we ask you to show the formulas and calculations in the ESI file. (Follow comment number 18)
Response: We corrected the calculation mistake in revised version and showed the formulas in the ESI file.
- In Line 257 to 261, the equation font need to be revised, and the proper citation needs to be referenced.
Response: The equation font has been revised in the revised manuscript, we also added relevant references.
- The characterization of the sample after the stability test needs to be added. (Fe-SEM and EDX with color mapping).
Response: The characterization of the sample after the stability test have been added and marked as Fig.S4 in the revised version.

Round 2
Reviewer 1 Report
Acceptable corrections
Reviewer 2 Report
All the comments are appended carefully.